# Understanding Compositional Data Augmentation in Typologically Diverse Morphological Inflection

**Farhan Samir    Miikka Silfverberg**
Natural Language Processing Group, University of British Columbia
`fsamir@cs.ubc.ca`

## Abstract

Data augmentation techniques are widely used in low-resource automatic morphological inflection to overcome data sparsity. However, the full implications of these techniques remain poorly understood. In this study, we aim to shed light on the theoretical aspects of the prominent data augmentation strategy STEM-CORRUPT (Silfverberg et al., 2017; Anastasopoulos and Neubig, 2019), a method that generates synthetic examples by randomly substituting stem characters in gold standard training examples. To begin, we conduct an information-theoretic analysis, arguing that STEMCORRUPT improves compositional generalization by eliminating spurious correlations between morphemes, specifically between the stem and the affixes. Our theoretical analysis further leads us to study the sample-efficiency with which STEMCORRUPT reduces these spurious correlations. Through evaluation across seven typologically distinct languages, we demonstrate that selecting a subset of datapoints with both high diversity *and* high predictive uncertainty significantly enhances the data-efficiency of STEMCORRUPT. However, we also explore the impact of typological features on the choice of the data selection strategy and find that languages incorporating a high degree of allomorphy and phonological alternations derive less benefit from synthetic examples with high uncertainty. We attribute this effect to phonotactic violations induced by STEMCORRUPT, emphasizing the need for further research to ensure optimal performance across the entire spectrum of natural language morphology.[1]

## 1   Introduction

Compositional mechanisms are widely believed to be the basis for human language production and comprehension (Baroni, 2020). These mechanisms involve the combination of simpler parts to form complex concepts, where valid combinations are licensed by a recursive grammar (Kratzer and Heim, 1998; Partee et al., 1984). However, domain general neural architectures often fail to generalize to new, unseen data in a compositional manner, revealing a failure in inferring the data-generating grammar (Kim and Linzen, 2020; Lake and Baroni, 2018; Wu et al., 2023). This failure hinders these models from closely approximating the productivity and systematicity of human language.

Consider the task of automatic morphological inflection, where models must learn the underlying rules of a language's morphoysyntax to produce the inflectional variants for any lexeme from a large lexicon. The task is challenging: the models must efficiently induce the rules with only a small human-annotated dataset. Indeed, a recent analysis by Goldman et al. (2022) demonstrates that even state-of-the-art, task-specific automatic inflection models fall short of a compositional solution: they perform well in random train-test splits, but struggle in compositional ones where they must inflect lexemes that were unseen at training time.

Nevertheless, there is reason for optimism. Several works have shown that automatic inflection models come much closer to a compositional solution when the human-annotated dataset is complimented by a synthetic data-augmentation procedure (Liu and Hulden, 2022; Silfverberg et al., 2017; Anastasopoulos and Neubig, 2019; Lane and Bird, 2020; Samir and Silfverberg, 2022), where morphological affixes are identified and attached to synthetic lexemes distinct from those in the training dataset (Fig. 2). However, little is understood about this prominent data augmentation method and the extent to which it can improve compositional generalization in neural word inflection. In this work, we seek to reveal the implicit assumptions about morpheme distributions made by this rule-based augmentation scheme, and analyze the

---

[1]Our code is available at https://github.com/smfsamir/understanding-augmentation-morphology.

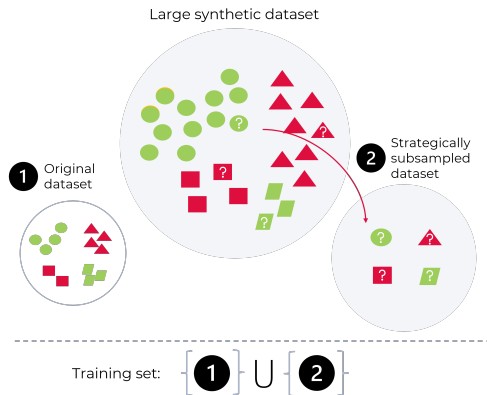

Figure 1: Given a small human-annotated dataset and a large pool of synthetic examples, we find that sampling a subset of data representing both diversity (a multitude of shapes) and high predictive uncertainty (shapes with a question mark) are on average more sample-efficient in improving compositional generalization in morphological inflection.

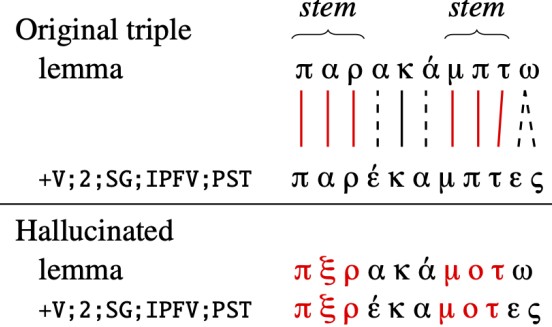

[Illustration from Anastasopoulos and Neubig (2019)]

Figure 2: STEMCORRUPT: a data augmentation method, where the stem – aligned subsequences of length 3 or greater in the input and output – is mutated by substitution with random characters from the alphabet.

effect of these assumptions on learning of cross-linguistically prevalent morphological patterns.

To this end, our work presents the first theoretical explanation for the effectiveness of compositional data augmentation in morphological inflection. Through an information-theoretic analysis (Section 3), we show that this method eliminates "spurious correlations" (Gardner et al., 2021) between the word's constituent morphemes, specifically between the stem (e.g., *walk*) and the inflectional affix (e.g., *-ed*). By removing these correlations, the training data distribution becomes aligned with concatenative morphology, where a word can be broken down into two independent substructures: the stem (identifying the lexeme) and the inflectional affixes (specifying grammatical function). This finding sheds light on why the method is widely attested to improve compositional generalization (Liu and Hulden, 2022), as concatenative morphological distributions are cross-linguistically prevalent (Haspelmath and Sims, 2013).

We go on to show, however, that the augmentation method tends towards removing *all* correlations between stems and affixes, whether spurious or not. Unfortunately, this crude representation of concatenative morphology, while reasonable in broad strokes, is violated in virtually all languages to varying degrees by long-distance phonological phenomena like vowel harmony and reduplication. Thus, our analysis demonstrates that while the method induces a useful approximation to concatenative morphology, there is still ample room for improvement in better handling of allomorphy and phonological alternations.

Building on our theoretical analysis, we in-

vestigate whether it is possible to improve the sample-efficiency with which the data augmentation method induces probabilistic independence between stems and affixes. Specifically, we investigate whether we can use a small subset of the synthetic data to add to our training dataset. We find that selecting a subset that incorporates both high predictive uncertainty and high diversity (see Fig. 1) is significantly more efficient in removing correlations between stems and affixes, providing an improvement in sample-efficiency for languages where the morphological system is largely concatenative. At the same time, in accordance with our theoretical analysis, this selection strategy impairs performance for languages where phonological alternations are common.

Our work contributes to a comprehensive understanding of a prominent data augmentation method from both a theoretical (Section 3) and practical standpoint (Section 4). Through our systematic analysis, we aim to inspire further research in the analysis and evaluation of existing compositional data augmentation methods (reviewed in Section 6), as well as the development of novel augmentation methods that can better capture cross-linguistic diversity in morphological patterns.

## 2 Preliminaries

In automatic morphological inflection, we assume access to a gold-standard dataset $D_{train}$ with triples of $\langle \mathbf{X}, \mathbf{Y}, \mathbf{T} \rangle$, where $\mathbf{X}$ is the character sequence of the lemma, $\mathbf{T}$ is a morphosyntactic description (MSD), and $\mathbf{Y}$ is the character sequence of the inflected form.[2] The goal is then to learn the distribution $P(\mathbf{Y}|\mathbf{X}, \mathbf{T})$ over inflected forms conditioned

---

[2]For example, <dog, dogs, N+PL>.

on a lemma and MSD .

**Generating a synthetic training dataset with STEMCORRUPT.** For many languages, $D_{train}$ is too small for models to learn and make systematic morphological generalizations. Previous work has found that generating a complementary synthetic dataset $D_{train}^{Syn}$ using a data augmentation technique can substantially improve generalization (Anastasopoulos and Neubig, 2019; Silfverberg et al., 2017, among others).

The technique, henceforth called STEMCOR-RUPT, works as follows: We identify the aligned subsequences (of length 3 or greater) between a lemma $\mathbf{X}$ and an inflected form $\mathbf{Y}$, which we denote the stem.[3] We then substitute some of the characters in the stem with random ones from the language's alphabet; Fig. 2. The STEMCORRUPT procedure has a hyperparameter $\theta$. It sets the probability that a character in the stem will be substituted by a random one; a higher value of $\theta$ (approaching 1) indicates a greater number of substitutions in the stem.[4]

How does STEMCORRUPT improve compositional generalization? Despite the widespread adoption of this technique for automatic morphological inflection and analysis, this fundamental question has heretofore remained unanswered. In the next section, we argue that STEMCORRUPT improves compositional generalization by removing correlations between inflectional affixes and the stems to which they are attached. By enforcing independence between these two substructures, STEMCORRUPT facilitates productive reuse of the inflectional affixes with other lexemes. That is, our theoretical argument shows that the effectiveness of the method arises from factoring the probability distribution to approximate concatenative morphology, a cross-linguistically prevalent strategy for word formation where words can be "neatly segmented into roots and affixes" (Haspelmath and Sims, 2013). We formalize and prove this in the following section.

## 3 STEMCORRUPT induces compositional structure

In this section, we analyze the ramifications of training on the synthetic training dataset generated by STEMCORRUPT ($D_{train}^{Syn}$) and the human-annotated training dataset ($D_{train}$). Our analysis

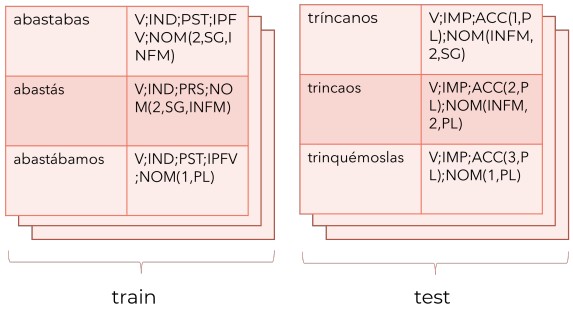

Figure 3: Models are trained and evaluated on entirely different lexemes.

focuses on asymptotic behaviour, specifically, how the conditional distribution $P(\mathbf{Y}|\mathbf{X}, \mathbf{T})$ evolves as we add more augmented data ($|D_{train}^{Syn}| \to \infty$).

**Theorem 1.** *For all $\langle \mathbf{X}, \mathbf{Y}, \mathbf{T} \rangle$ datapoints in $D_{train}$, assume that $\mathbf{X}$ and $\mathbf{Y}$ share a stem $\mathbf{Y}_{stem}$ that is non-empty, and let $\mathbf{Y}_{affix}$ be the remaining characters of $\mathbf{Y}$. Let $\mathbf{X}_{stem}$ and $\mathbf{X}_{affix}$ be analogously defined. Further, let $D_{train}^{Syn}$ be generated with STEMCORRUPT using $\theta = 1$.[5] Next, consider the data-generating probability distribution $P(\mathbf{Y}|\mathbf{X}, \mathbf{T})$ over $D_{train}^{Syn} \cup D_{train}$. Then, as $|D_{train}^{Syn}| \to \infty$, we have that $P(\mathbf{Y}|\mathbf{X}, \mathbf{T}) \equiv P(\mathbf{Y}_{stem}, \mathbf{Y}_{affix}|\mathbf{X}, \mathbf{T}) = P(\mathbf{Y}_{affix}|\mathbf{X}_{affix}, \mathbf{T})P(\mathbf{Y}_{stem}|\mathbf{X}_{stem}).$*

*Remark* 1 (*Concatenative compositionality*). The augmentation method thus makes the model more effective at capturing concatenative morphological patterns, as the conditional probability distribution becomes factorized into a root generation component ($P(\mathbf{Y}_{stem}|\cdot)$) and an affix generation component ($P(\mathbf{Y}_{affix}|\cdot)$). Crucially, this removes the potential for any spurious correlation between these two substructures.[6]

*Remark* 2 (*Stem-affix dependencies*). While concatenation is a cross-linguistically prevalent strategy for inflection (Haspelmath and Sims, 2013), stems and affixes are rarely entirely independent. Therefore, enforcing complete independence between these structures is an overly strong constraint that STEMCORRUPT places on the training data. The constraint is consistently violated in Turkish, for example, where front-back vowel harmony constraints dictate that the vowels in the suffix share the same front/back feature as the initial vowel in the stem. This leads to forms like "daların" and pre-

---

[3]The stem can thus be discontinuous; see Fig. 2.

[4]We use the implementation here, which sets $\theta = 0.5$.

[5]That is, we substitute *all* characters in the stem.

[6]Additionally, this factorization is likely to reducing overfitting: The model effectively learns to minimize the negative log-likelihood of this simplified and factorized data distribution $P(\mathbf{Y}_{stem}|\cdot)P(\mathbf{Y}_{affix}|\cdot)$, rather than the complex joint distribution $P(\mathbf{Y}_{stem}, \mathbf{Y}_{affix}|\cdot)$.

vents forms like "dalerin", "dalerın", or "dalarin" (Kabak, 2011). In Section 5, we show that STEM-CORRUPT regularly generates examples violating vowel harmony, and that this can undermine its effectiveness. Nevertheless, the empirical success of STEMCORRUPT demonstrates that the benefits of its concatenative bias outweigh its limitations.

*Remark* 3 (*Comparison to previous accounts*). Our analysis provides a simple yet the most accurate characterization of STEMCORRUPT. Previous works have called STEMCORRUPT a beneficial "copying bias " (Liu and Hulden, 2022; Jaidi et al., 2022) or a strategy for mitigating overgeneration of common character n-grams (Anastasopoulos and Neubig, 2019). However, our analysis demonstrates that neither of these characterizations are entirely accurate. First, the denotation of a "copying bias" is only suggestive of the second factor in our statement $P(\mathbf{Y}_{stem}|\mathbf{X}_{stem})$, and does not address the impact on affix generation. In contrast, our analysis shows that both stem and affix generation are affected. Furthermore, alleviating overfitting to common character sequences is also misleading, as it would suggest that STEMCORRUPT serves the same purpose as standard regularization techniques like label smoothing (Müller et al., 2019).[7]

## 3.1 Proving the theorem

The proof of the theorem is straightforward with the following proposition (proved in Appendix B).

**Proposition 1.** *As $|D_{train}^{Syn}| \to \infty$, the mutual information between certain pairs of random variables declines:*

*(i)* $I(\mathbf{Y}_{stem}; \mathbf{T}) \to 0$

*(ii)* $I(\mathbf{Y}_{stem}; \mathbf{X}_{affix}) \to 0$

*(iii)* $I(\mathbf{Y}_{affix}; \mathbf{Y}_{stem}) \to 0$

*(iv)* $I(\mathbf{Y}_{affix}; \mathbf{X}_{stem}) \to 0$

*Proof of Theorem 1.* By the definition of $\mathbf{Y} = \mathbf{Y}_{stem}\mathbf{Y}_{affix}$, we have that $P(\mathbf{Y}|\mathbf{X}, \mathbf{T}) \equiv P(\mathbf{Y}_{stem}, \mathbf{Y}_{affix}|\mathbf{X}, \mathbf{T})$. Then, by the chain rule of probability, we have $P(\mathbf{Y}_{affix}|\mathbf{Y}_{stem}, \mathbf{X}, \mathbf{T})P(\mathbf{Y}_{stem}|\mathbf{X}, \mathbf{T})$. We first deconstruct the second factor. By Proposition 1 (i), we have that the second factor $P(\mathbf{Y}_{stem}|\mathbf{X}, \mathbf{T}) = P(\mathbf{Y}_{stem}|\mathbf{X})$, since the

stem is invariant with respect to the inflectional features. Then, by Proposition 1 (ii), we have that $P(\mathbf{Y}_{stem}|\mathbf{X}) = P(\mathbf{Y}_{stem}|\mathbf{X}_{stem}, \mathbf{X}_{affix}) = P(\mathbf{Y}_{stem}|\mathbf{X}_{stem})$, since the stem is invariant with respect to the inflectional affix of the lemma.

Next, we tackle the factor $P(\mathbf{Y}_{affix}|\mathbf{Y}_{stem}, \mathbf{X}, \mathbf{T})$. By parts (iii) and (iv) of Proposition 1, we have that this can be simplified to $P(\mathbf{Y}_{affix}|\mathbf{X}_{affix}, \mathbf{T})$. Taken together, we have that $P(\mathbf{Y}|\mathbf{X}, \mathbf{T})$ can be decomposed into $P(\mathbf{Y}_{affix}|\mathbf{X}_{affix}, \mathbf{T})P(\mathbf{Y}_{stem}|\mathbf{X}_{stem})$. □

**Investigating STEMCORRUPT's sample efficiency**. So far, we have studied the behaviour of STEMCORRUPT through an asymptotic argument, demonstrating that in the infinite limit of $|D_{train}^{Syn}|$, STEMCORRUPT enforces complete independence between stems and affixes. In doing so, it likely removes a number of spurious correlations between stems and affixes, thus providing a theoretical explanation for its attested benefit in improving compositional generalization. However, our theoretical analysis, while informative of the overall effect of STEMCORRUPT, says little about the sample efficiency of the method in practice.

Indeed, recent studies in semantic parsing have demonstrated that sample efficiency can be greatly increased by strategically sampling data to overcome spurious correlations that hinder compositional generalization in non-IID data splits (Oren et al., 2021; Bogin et al., 2022; Gupta et al., 2022). In the following section, we examine whether strategic data selection can yield similar benefits in the context of typologically diverse morphological inflection.

## 4 Extracting sample-efficient training sets from STEMCORRUPT

**Problem setup**. We use the following problem setup from Oren et al. (2021) to investigate whether the sample-efficiency of STEMCORRUPT can be improved. Recall that we have a dataset $D_{train}$ with gold triples of $\langle \mathbf{X}, \mathbf{Y}, \mathbf{T} \rangle$. Further, we have a synthesized dataset $D_{train}^{Syn}$ where $|D_{train}^{Syn}| \gg |D_{train}|$. Our goal is now to select $\hat{D}_{train}^{Syn} \subset D_{train}^{Syn}$ so that training the model on $D_{train} \cup \hat{D}_{train}^{Syn}$ maximizes performance on a held-out compositional testing split.

## 4.1 Model and training

We start by training an inflection model $\mathcal{M}$ on the gold-standard training data, denoted as $D_{train}$. Fol-

---

[7]Our preliminary experiments did not support this position; compositional generalization performance was not sensitive to label smoothing, yet was significantly improved by STEMCORRUPT.

lowing Wu et al. (2021); Liu and Hulden (2022), we employ Transformer (Vaswani et al., 2017) for $\mathcal{M}$. We use the fairseq package (Ott et al., 2019) for training our models and list our hyperparameter settings in Appendix A. We conduct all of our experiments with $|D_{train}| = 100$ gold-standard examples, adhering to the the low-resource setting for SIGMORPHON 2018 shared task for each language. We next describe the construction of $\hat{D}_{train}^{Syn}$.

## 4.2 Subset sampling strategies

Here, we introduce a series of strategies for sampling from $D_{train}^{Syn}$ oriented for improving compositional generalization. Broadly, we focus on selecting subsets that reflect either high structural diversity, high predictive uncertainty, or both, as these properties have been tied to improvements in compositional generalization in prior work on semantic parsing (e.g., Bogin et al., 2022), an NLP research area where compositionality is well studied.

**RANDOM.** Our baseline sampling method is to construct $\hat{D}_{train}^{Syn}$ by sampling from the large synthetic training data $D_{train}^{Syn}$ uniformly.

**UNIFORM MORPHOLOGICAL TEMPLATE (UMT).** With this method, we seek to improve the structural diversity in our subsampled in synthetic training dataset $\hat{D}_{train}^{Syn}$. Training on diverse subset is crucial, as the SIGMORPHON 2018 shared task dataset is imbalanced in frequency of different morphosyntactic descriptions (MSDs).[8] These imbalances can pose challenges to the model in generalizing to rarer MSDs. To incorporate greater structural diversity, we employ the templatic sampling process proposed by Oren et al. (2021). Specifically, we modify the distribution over MSDs to be closer to uniform in $\hat{D}_{train}^{Syn}$.

Formally, we sample without replacement from the following distribution: $q_\alpha(\mathbf{X}, \mathbf{Y}, \mathbf{T}) = p(\mathbf{T})^\alpha / \sum_{\mathbf{T}} p(\mathbf{T})^\alpha$ where $p(\mathbf{T})$ is the proportion of times that $\mathbf{T}$ appears in $D_{train}^{Syn}$. We consider two cases: $\alpha = 0$ corresponds to sampling MSDs from a **uniform** distribution (**UMT**), while $\alpha = 1$ corresponds to sampling tags according to the **empirical** distribution over MSDs (**EMT**).

---

[8]For example, the low-resource Georgian training dataset contains 9 instances of of nouns inflected for PL;ERG, but only one instance of a noun inflected for SG;ERG.

**HIGHLOSS.** Next, we employ a selection strategy that selects datapoints that have high predictive uncertainty to the initial model $\mathcal{M}$. Spurious correlations between substructures (like $\mathbf{Y}_{stem}$ and $\mathbf{T}$; Section 3) will exist in any dataset of bounded size (Gupta et al., 2022; Gardner et al., 2021), and we conjecture that selecting high uncertainty datapoints will efficiently mitigate these correlations.

We quantify the uncertainty of a synthetic datapoint in $D_{train}^{Syn}$ by computing the negative log-likelihood (averaged over all tokens in the the target $\mathbf{Y}$) for each synthetic datapoint in $D_{train}^{Syn}$. Next, we select the synthetic datapoints with the highest uncertainty and add them to $\hat{D}_{train}^{Syn}$.

To thoroughly demonstrate that incorporating predictive uncertainty is important for yielding training examples that counteract the spurious dependencies in the ground-truth training dataset, we benchmark it against another subset selection strategy **LOWLOSS**. With this method, we instead select synthetic datapoints that the model finds easy, i.e., those with the lowest uncertainty scores. We hypothesize this strategy will yield less performant synthetic training datasets, as it is biased towards selecting datapoints that corroborate rather than counteract the spurious correlations learned by $\mathcal{M}$.

**UMT/EMT+ LOSS.** Finally, we test a hybrid approach containing both high structural diversity and predictive uncertainty by combining UMT/EMT and HIGHLOSS. First, we sample an MSD $\mathbf{T}$ (according to the MSD distribution defined by UMT/EMT) and then select the most uncertain synthetic datapoint for that $\mathbf{T}$.

## 5 Experiments and Results

**Data**. We use data from the UniMorph project (Batsuren et al., 2022), considering typological diversity when selecting languages to include. We aim for an evaluation similar in scope to Muradoglu and Hulden (2022). That is, broadly, we attempt to include types of languages that exhibit variation in inflectional characteristics such as inflectional synthesis of the verb, exponence, and morphological paradigm size (Haspelmath et al., 2005). Our selected languages can be seen in Fig. 4. We provide further information on the languages in Appendix C.

**Obtaining a large synthetic dataset**. In order to

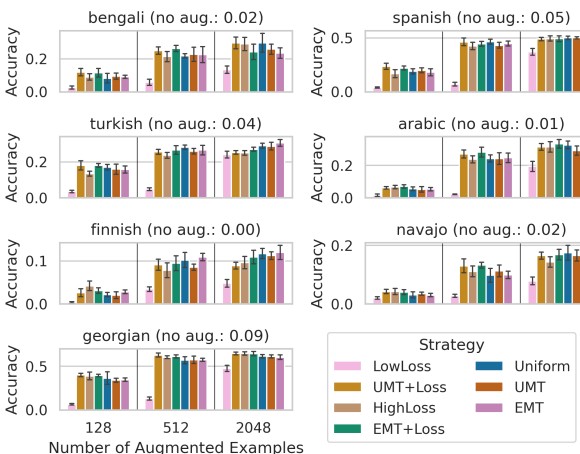

Figure 4: Performance after training on $D_{train} \cup \hat{D}_{train}^{Syn}$, for varying sizes of $\hat{D}_{train}^{Syn}$. In the subtitle for each language, we also list the performance from training on no augmented data in parentheses ($\hat{D}_{train}^{Syn} = \emptyset$).

generate the large augmentation dataset $D_{train}^{Syn}$ for every language, we generate $10,000$ augmented datapoints for every language by applying STEM-CORRUPT to their respective *low* datasets from SIGMORPHON 2018 (Cotterell et al., 2018).

**Generating a compositional generalization test set**. For generating test sets, we adopt the lemma-split approach of Goldman et al. (2022). Specifically, we use all available data from SIGMORPHON2018 for the target language, excluding any lexemes from the *low* setting since those were used to train the initial model $\mathcal{M}$ (Section 4.1). The remaining lexemes and their associated paradigms comprise our compositional generalization test set; see Fig. 3.

**Populating** $\hat{D}_{train}^{Syn}$. We evaluate the performance of all methods listed in Section 4 in selecting $\hat{D}_{train}^{Syn}$. We evaluated the performance of using $\hat{D}_{train}^{Syn}$ of sizes ranging from 128 to 2048 examples, increasing the size by powers of 2.

### 5.1 Results

We demonstrate the results of each strategy for all languages, considering each combination of language, subset selection strategy, and $|D_{train}^{Syn}|$, thus obtaining 35 sets of results. For each setting, we report the performance achieved over 6 different random initializations, along with their respective standard deviations. For brevity, we show the results for $|\hat{D}_{train}^{Syn}| \in \{128, 512, 2048\}$; we include the expanded set of results (including $\{256, 1024\}$) in Appendix D.

**STEMCORRUPT improves compositional gener-**

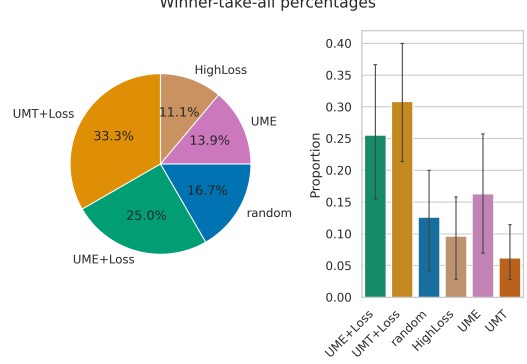

Figure 5: Summary of subset selection strategies performances from. Left: percentage of times each strategy gets the best performance out of 35 settings (across each of the 7 languages and 5 $\hat{D}_{train}^{Syn}$ sizes). Right: bootstrapped confidence intervals for the percentages on the left.

**alization.** At a high level, we find that data augmentation brings substantial benefits for compositional generalization compared to models trained solely on the gold-standard training data $D_{train}$. Without STEMCORRUPT, the initial model $\mathcal{M}$ for every language achieves only single-digit accuracy, while their augmented counterparts perform significantly better. For instance, the best models for Georgian and Spanish achieve over 50% accuracy. These findings agree with those of Liu and Hulden (2022) who found that unaugmented Transformer models fail to generalize inflection patterns.

We also find that performance tends to increase as we add more synthetic data; the best models for every language are on the higher end of the $|\hat{D}_{train}^{Syn}|$ sizes. This finding agrees with our theoretical results that the dependence between the stem ($\mathbf{Y}_{stem}$) and that of the inflectional affix ($\mathbf{Y}_{affix}$) is weakened as we add more samples from STEMCORRUPT (Section 3; Proposition 1).

**Effective subsets have high diversity *and* predictive uncertainty**. Our analysis reveals statistically significant differences between the subset selection strategies, highlighting the effectiveness of the hybrid approaches (UMT/EMT+LOSS) that consider both diversity and predictive uncertainty. Among the strategies tested, the UMT+LOSS method outperformed all others in approximately one-third of the 35 settings examined, as indicated in Figure 5 (left). The improvements achieved by the UMT+LOSS method over a random baseline were statistically significant ($p < 0.05$) according to a bootstrap percentile test (Efron and Tibshirani, 1994), as shown in Figure 5 (right). Moreover, our results also show that the EMT+LOSS strategy closely followed the UMT+LOSS approach,

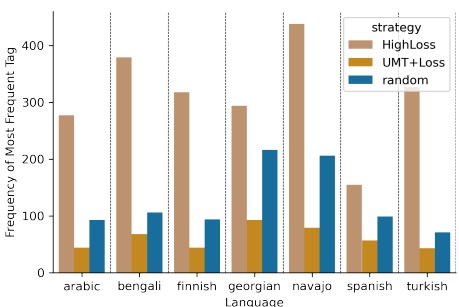

Figure 6: The frequency of the most commonly sampled morphosyntactic description by three of subset selection methods.

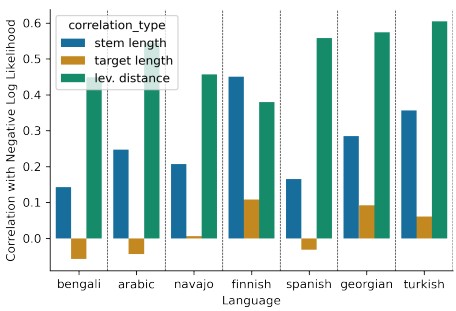

Figure 7: Pearson correlations between Negative Log Likelihood and three other metrics: length of the stem, length of the target inflected form, and Levenshtein distance between the ground-truth target form and the augmented target.

achieving the highest performance in a quarter of the cases. In contrast, the same strategies without the uncertainty component were much less effective. For instance, UMT never achieved the best performance in any combination of the languages and $|\hat{D}^{Syn}_{train}|$ sizes, highlighting that selecting a diverse subset without factoring in predictive uncertainty is suboptimal.

Furthermore, selecting datapoints based solely on high predictive uncertainty without considering diversity (HIGHLOSS) is an ineffective strategy, having the second lowest proportion of wins (Fig. 5, right). Empirically, we find this may be attributed to the HIGHLOSS strategy fixating on a particular MSD, as shown in Fig. 6, rather than exploring the full distribution of MSDs. The figure displays the frequency of the most commonly sampled morphosyntactic description for each of UMT+LOSS, RANDOM, and HIGHLOSS strategies. Across all languages, the HIGHLOSS method samples the most frequent tag much more often than the RANDOM and UMT+LOSS methods.[9]

**"Easy" synthetic datapoints have low sample efficiency.** As hypothesized, the datapoints with low uncertainty hurt performance. We attribute this to the LowLoss strategy selecting datapoints with a smaller number of substitutions to the stem. In Fig. 7, we show that the number of edits made to the stem – as measured by Levenshtein distance between the corrupted target sequence and the uncorrupted version – is strongly correlated with the uncertainty assigned to the synthetic datapoint across all languages. Moreover, the correlation between the number of edits and uncertainty

is higher than the correlation with other plausible factors driving uncertainty, namely stem length and target length.[10] Overall, the lagging sample efficiency of LOWLOSS corroborates our theory; STEMCORRUPT is effective because it generates datapoints where the stem has no correspondence with the affix. LOWLOSS counteracts its effectiveness, as it is biased towards datapoints where spurious dependencies between the stem and affix are maintained.[11]

**Selecting by high predictive uncertainty worsens performance when there are stem-affix dependencies.** We found that the UMT+LOSS strategy improves performance for 5 out of 7 languages compared to the RANDOM baseline. The improvement ranges from 4.8 (Georgian) to small declines of $-1.9$ (Turkish) and $-0.9$ (Finnish). The declines for Finnish and Turkish are partly due to a mismatch between the generated synthetic examples and the languages' morphophonology. STEMCORRUPT will generate synthetic examples that violate vowel harmony constraints between the stem and the affix. For instance, it may replace a front vowel in the stem with a back one. As a result, UMT+LOSS will select such harmony-violating examples more often, since they have greater uncertainty to the initial model $\mathcal{M}$ (Section 4.1), resulting in the augmented model tending to violate the harmony restrictions more often. Indeed, for Turkish, the average uncertainty for synthetic examples violating vowel harmony (0.46) is significantly higher than those that adhere to vowel harmony

---

[9]The reason that uncertainty estimates are higher for a given MSD is not entirely clear. In our investigation, we found a small correlation ($\rho = 0.15$) between the morphosyntactic description frequency and uncertainty. However, there are likely other factors beyond frequency that contribute to higher uncertainty; for example, morphological fusion (Bickel and Nichols, 2013; Rathi et al., 2021).

[10]Target length is known to contribute to predictive uncertainty (Eikema and Aziz, 2020; Kim and Linzen, 2020), and stem length is a confounding factor of the levenshtein distance calculation.

[11]The ineffectiveness of LOWLOSS also corroborates that STEMCORRUPT does not simply induce a "copying bias" (Remark 3), since then we would expect all of the subset selection methods to perform similarly.

| Language | Improvement |
|----------|-------------|
| Georgian | 4.8 |
| Bengali | 2.5 |
| Spanish | 1.3 |
| Navajo | 1.1 |
| Arabic | 0.4 |
| Finnish | −0.9 |
| Turkish | −1.9 |

Table 1: Improvement of UMT+Loss relative to the random baseline, averaged over all possible sizes of $\hat{D}_{train}^{Syn}$.

(0.39), as assessed by a bootstrap percentile test ($p < 0.05$). This finding also corroborates our theory from Section 3: STEMCORRUPT eliminates dependencies between stems and affixes, even when the dependencies are real rather than spurious. This shortcoming of STEMCORRUPT is exacerbated by selecting examples with high uncertainty, as these examples are less likely to adhere to stem-affix constraints like vowel harmony.

**Takeaways.** Aligning with the semantic parsing literature on efficient compositional data augmentation (Oren et al., 2021; Bogin et al., 2022; Gupta et al., 2022), we find that certain subsets of data are on average more efficient at eliminating spurious correlations between substructures – in the case of morphological inflection, the relevant substructures being the individual morphemes: the stem ($\mathbf{Y}_{stem}$) and affix ($\mathbf{Y}_{affix}$; Section 3). However, the sample-efficiency gains from strategic sampling are less dramatic than in semantic parsing (see, for example, Oren et al., 2021).

We provided empirical evidence that the gains are tempered by STEMCORRUPT's tendencies to violate stem-affix constraints in its synthetic training examples, such as vowel harmony constraints in Turkish. Thus, further work is needed to adapt or supplant STEMCORRUPT for languages where such long-range dependencies are commonplace. In doing so, the data selection strategies are likely to fetch greater gains in sample efficiency.

## 6 Related work

**Compositional data augmentation methods.** In general, such methods synthesize new data points by splicing or swapping small parts from existing training data, leveraging the fact that certain constituents can be interchanged while preserving overall meaning or syntactic structure. A common approach is to swap spans between pairs of datapoints when their surrounding contexts are identical (Andreas, 2020; Guo et al., 2020; Jia and Liang, 2016, inter alia). Recently, Akyürek et al. (2021) extended this approach, eschewing rule-based splic-

ing in favour of neural network-based recombination. Chen et al. (2023) review more compositional data augmentation techniques, situating them within the broader landscape of limited-data learning techniques for NLP.

**Extracting high-value subsets in NLP training data.** Oren et al. (2021); Bogin et al. (2022); Gupta et al. (2022) propose methods for extracting diverse sets of abstract templates to improve compositional generalization in semantic parsing. Muradoglu and Hulden (2022) train a baseline model on a small amount of data and use entropy estimates to select new data points for annotation, reducing annotation costs for morphological inflection. Swayamdipta et al. (2020) identify effective data subsets for training high-quality models for question answering, finding that small subsets of ambiguous examples perform better than randomly selected ones. Our work is also highly related to active-learning in NLP (Tamkin et al., 2022; Yuan et al., 2020; Margatina et al., 2021, inter alia); however we focus on selecting synthetic rather than unlabeled datapoints, and our experiments are geared towards compositional generalization rather than IID performance.

**Compositional data splits in morphological inflection.** Assessing the generalization capacity of morphological inflections has proven a challenging and multifaceted problem. Relying on standard "IID" (Oren et al., 2021; Liu and Hulden, 2022) splits obfuscated (at least) two different manners in which inflection models fail to generalize compositionally.

First, Goldman et al. (2022) uncovered that generalizing to novel lexemes was challenging for even state of the art inflection models. Experiments by Liu and Hulden (2022) however showed that the STEMCORRUPT method could significantly improve generalization to novel lexemes. Our work builds on theirs by contributing to understanding the relationship between STEMCORRUPT and lexical compositional generalization. Specifically, we studied the structure of the probability distribution that StemCorrupt promotes (Section 3), and the conditions under which it succeeds (Remark 1) and fails (Remark 2).

Second, Kodner et al. (2023) showed that inflection models also fail to generalize compositionally to novel feature combinations, even with agglutinative languages that have typically have a strong one-to-one alignment between morphological features and affixes. Discovering strategies to facilitate com-

positional generalization in terms of novel feature combinations remains an open-area of research.

# 7 Conclusion

This paper presents a novel theoretical explanation for the effectiveness of STEMCORRUPT, a widely-used data augmentation method, in enhancing compositional generalization in automatic morphological inflection. By applying information-theoretic constructs, we prove that the augmented examples work to improve compositionality by eliminating dependencies between substructures in words – stems and affixes. Building off of our theoretical analysis, we present the first exploration of whether the sample efficiency of reducing these spurious dependencies can be improved. Our results show that improved sample efficiency is achievable by selecting subsets of synthetic data reflecting high structural diversity and predictive uncertainty, but there is room for improvement – both in strategic sampling strategies and more cross-linguistically effective data augmentation strategies that can represent long distance phonological alternations.

Overall, NLP data augmentation strategies are poorly understood (Dao et al., 2019; Feng et al., 2021) and our work contributes to filling in this gap. Through our theoretical and empirical analyses, we provide insights that can inform future research on the effectiveness of data augmentation methods in improving compositional generalization.

# 8 Limitations

**Theoretical analysis.** We make some simplifying assumptions to facilitate our analysis in Section 3. First, we assume that the stem between a gold-standard lemma and inflected form $\mathbf{X}$ and $\mathbf{Y}$ is discoverable. This is not always the case; for example, with suppletive inflected forms, the relationship between the source lemma and the target form is not systematic. Second, we assume that all characters in the stem are randomly substituted, corresponding to setting the $\theta = 1$ for STEMCORRUPT. This does not correspond to how we deploy STEMCORRUPT; the implementation provided by Anastasopoulos and Neubig (2019) sets $\theta = 0.5$ and we use this value for our empirical analysis Section 5. We believe the analysis under $\theta = 1$ provides a valuable and accurate characterization of STEMCORRUPT nonetheless and can be readily extended to accommodate the $0 < \theta < 1$ case in future work.

**Empirical analysis.** In our empirical analysis, we acknowledge two limitations that can be addressed in future research. First, we conduct our experiments using data collected for the SIGMORPHON 2018 shared task, which may not contain a naturalistic distribution of morphosyntactic descriptions since it was from an online database (Wiktionary). In future work, we aim to replicate our work in a more natural setting such as applications for endangered language documentation (Muradoglu and Hulden, 2022; Moeller et al., 2020), where the morphosyntactic description distribution is likely to be more imbalanced. Second, we perform our analyses in an extremely data-constrained setting where only 100 gold-standard examples are available. In higher resourced settings, data augmentation with STEMCORRUPT may provide a much more limited improvement to compositional generalization; indeed the compositional generalization study of morphological inflection systems by Goldman et al. (2022) demonstrates that the disparity between IID generalization and compositional generalization largely dissipates when the model is trained on more gold-standard data.

# 9 Acknowledgements

We thank the EMNLP reviewers, the area chair, Vered Shwartz, Kat Vylomova, and Adam Wiemerslage for helpful discussion and feedback on the manuscript. We also thank Ronak D. Mehta for insightful discussion on properties of mutual information. This work was supported by an NSERC PGS-D scholarship to the first author.

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

## A Transformer training details

| | |
|---|---:|
| Batch size | 16 |
| Label smoothing | 0.2 |
| Warmup updates | 4000 |
| Total updates | 6000 |
| Dropout | 0.3 |
| Encoder layers | 4 |
| Decoder layers | 4 |
| Attention dropout | 0.1 |
| Adam $(\beta_1, \beta_2)$ | $(0.9, 0.999)$ |

## B Proof of Proposition 1

We recall proposition 1:

**Proposition 1.** *As $|D_{train}^{Syn}| \to \infty$, the mutual information between certain pairs of random variables declines:*

(i) $I(\mathbf{Y}_{stem}; \mathbf{T}) \to 0$

(ii) $I(\mathbf{Y}_{stem}; \mathbf{X}_{affix}) \to 0$

(iii) $I(\mathbf{Y}_{affix}; \mathbf{Y}_{stem}) \to 0$

(iv) $I(\mathbf{Y}_{affix}; \mathbf{X}_{stem}) \to 0$

Our proof hinges on the fact that mutual information $I(X;Y)$ is convex in the conditional distribution $P(Y|X)$ when the marginal distribution $P(X)$ is held constant, due to Thomas and Cover (2006).

**Theorem 2** (Thomas & Cover). *Let $(X,Y) \sim p(x,y) = p(x)p(y|x)$. The mutual information $I(X;Y)$ is a concave function of $p(x)$ for fixed $p(y|x)$ and a convex function of $p(y|x)$ for fixed $p(x)$.*

This theorem is useful for our argument since the data augmentation algorithm results in the marginal distribution over some random variables being affected (namely $\mathbf{Y}_{stem}, \mathbf{X}_{stem}$) and other marginals staying fixed ($\mathbf{T}, \mathbf{X}_{affix}, \mathbf{Y}_{affix}$). This enables us to invoke the latter half of the theorem ("convex function of $p(y|x)$ for fixed $p(x)$") and thus obtain an upper bound on the mutual information between the pairs of variables stated in proposition 1. We will argue that this upper bound will decline to 0 as we take $|D_{train}^{Syn}| \to \infty$, and thus the mutual information must also decline to 0.

*Proof.* Let $I_G := I(\mathbf{T}; \mathbf{Y}_{stem})$ be the mutual information between the random variables $\mathbf{T}$ and $\mathbf{Y}_{stem}$ in the human annotated dataset $D_{train}$, where $\mathbf{T}$ is generated from some distribution $P(\mathbf{T})$ and $Y_{stem}$ be generated from $P_G(\mathbf{Y}_{stem}|\mathbf{T})$.

| | |
|---|---|
| Bengali | Indo-Aryan; 300M |
| Finnish | Uralic; 5.8M |
| Arabic | Semitic; 360M |
| Navajo | Athabaskan; 170K |
| Turkish | Turkic; 88M |
| Spanish | Indo-European; 592M |
| Georgian | Kartvelian; 3.7M |

Table 2: Languages assessed in our experiments on assessing the sample efficiency of data augmentation. We also list their language families and number of speakers.

Let $I_A := I(\mathbf{T}; \mathbf{Y}_{stem})$ be the mutual information between the random variables $\mathbf{T}$ and $\mathbf{Y}_{stem}$ in the synthetic dataset $D_{train}^{Syn}$, where $\mathbf{T}$ is generated from $P(\mathbf{T})$ (as before) and $\mathbf{Y}_{stem}$ is generated from $P_A(\mathbf{Y}_{stem}|\mathbf{T})$. The data augmentation algorithm generates the stem characters by uniformly sampling characters the a language's alphabet. Crucially, this means the mutual information $I_A = 0$, since the value of $\mathbf{Y}_{stem}$ is independent of the value of $\mathbf{T}$.

Then, let $I := I(\mathbf{T}; \mathbf{Y}_{stem})$ be the mutual information between the random variables $\mathbf{T}$ and $\mathbf{Y}_{stem}$ over $D_{train} \cup D_{train}^{Syn}$, where $(\mathbf{T}, \mathbf{Y}_{stem}) \sim (p(\mathbf{T}), \lambda P_G(\mathbf{Y}_{stem}|\mathbf{T}) + (1-\lambda)P_A(\mathbf{Y}_{stem}|T))$ and $\lambda := |D_{train}|/|D_{train} \cup D_{train}^{Syn}|$. By the convexity of mutual information (Theorem 2), we have that $I \leq \lambda I_G + (1-\lambda)I_A$.

As we take $|D_{train}^{Syn}| \to \infty$, we have that $\lambda I_G + (1-\lambda)I_A \to 0 \cdot I_G + 1 \cdot I_A = 0$. Thus, $I$ is lower bounded by zero (since mutual information is non-negative) and upper bounded by 0 as $D_{train}^{Syn} \to 0$ (by the above argument). Thus, we have that $I \to 0$, as desired. This proves (1).

The same argument can be applied to prove (ii), (iii), and (iv). For (ii), we let $\mathbf{X}_{affix}$ take the place of $\mathbf{T}$ and repeat the argument above. For (iii), we let $\mathbf{Y}_{affix}$ take the place of $\mathbf{T}$. For (iv), we let $\mathbf{Y}_{affix}$ take the place of $\mathbf{T}$ and let $\mathbf{X}_{stem}$ take the place of $\mathbf{Y}_{stem}$.

$\square$

## C Language information

In Table 2, we list the languages assessed in our experiments on assessing the sample efficiency of STEMCORRUPT with their language families and estimated number of speakers (Lewis, 2009).

## D Expanded results for assessing STEMCORRUPT's sample efficiency

Here we present the expanded set of results for Section 5; see Fig. 8. The results are the same as those in Fig. 4, except they also include $|\hat{D}_{train}^{Syn}| \in$

$\{256, 1024\}$ in addition to $\{128, 512, 2048\}$.

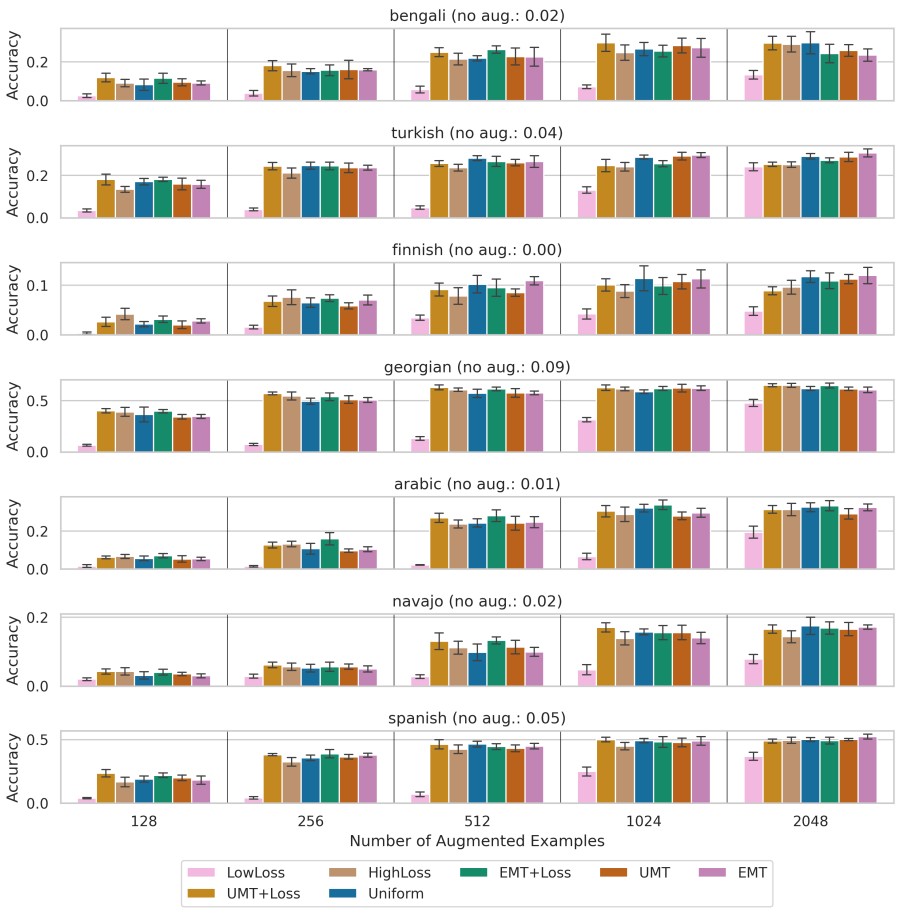

Figure 8: Performance after training on $D_{train} \cup \hat{D}_{train}^{Syn}$, for varying sizes of $\hat{D}_{train}^{Syn}$. In the subtitle for each language, we also list the performance from training on no augmented data in parentheses ($\hat{D}_{train}^{Syn} = \emptyset$).