# OpenReview forum: "Understanding Compositional Data Augmentation in Typologically Diverse Morphological Inflection"
_EMNLP/2023/Conference — EMNLP 2023 Main_

### Official Review · Reviewer_CMbW · 2023-08-03

**Soundness:** 4

**Excitement:**

4: Strong: This paper deepens the understanding of some phenomenon or lowers the barriers to an existing research direction.

**Paper Topic And Main Contributions:**

The paper proposed to better understand why the popular data-augmentation strategy employed for learning morphology inflection, is particularly effective. Specifically, they look at the STEMCORRUPT approach where affixes are automatically identified and attached to synthetic lexemes (unseen in the training data).
By using information-theoretic analysis they find that this data augmentation esp. helps removing spurious correlations between the stem and its affixes.
Within data augmentation, they introduce new sampling strategies based on uncertainty, diversity, to check which strategy helps in selecting the best subset that leads to test performance improvement.

**Questions For The Authors:**

1. One thing is unclear -- in Section 4.2 the authors introduce techniques to sample data, but I didn't quite understand how that compares with what existing approach STEMCORRECT does? Are these novel techniques or were these also used by STEMCORRECT?

**Reasons To Accept:**

1. The paper presents a nice study into understanding the effectiveness of an existing strategy. I have not many papers focussed on that. Its good to see the effort.

2. The paper is very clearly written and easy to understand.

3. The authors perform extensive experimentation across 7 languages and especially report results averaged across 6 random runs. The results are well in line with what the authors prove in Section 3 that under infinite data the distribution of stems and affix are independent of each other.

**Reasons To Reject:**

I don't have any major reasons, apart from the question below.

**Reproducibility:**

4: Could mostly reproduce the results, but there may be some variation because of sample variance or minor variations in their interpretation of the protocol or method.

**Reviewer Confidence:**

3: Pretty sure, but there's a chance I missed something. Although I have a good feel for this area in general, I did not carefully check the paper's details, e.g., the math, experimental design, or novelty.

---

> ### Author Rebuttal · Authors · 2023-08-29
>
> We agree with the reviewer’s assessment of the main contributions of the paper. We are pleased that the reviewer thinks that the theoretical analysis is concordant with the results of the empirical analyses; that the reviewer thinks the paper is well written and easy to understand; and that they find the analysis of an existing, effective strategy novel and valuable.
>
> > 1. One thing is unclear -- in Section 4.2 the authors introduce techniques to sample data, but I didn't quite understand how that compares with what existing approach STEMCORRECT does? Are these novel techniques or were these also used by STEMCORRECT?
>
> Employing sampling techniques in conjunction with StemCorrupt is a novel contribution of our work. We will clarify this in the camera-ready.

---

### Official Review · Reviewer_AGZF · 2023-08-04

**Soundness:** 3

**Excitement:**

4: Strong: This paper deepens the understanding of some phenomenon or lowers the barriers to an existing research direction.

**Missing References:**

- As mentioned in the **Reseson To Reject** section, it has been shown that MSD based splits reveal more about the performance and ability to generalize to unseen data of morphological inflection systems, as in this work: https://aclanthology.org/2023.acl-long.335.pdf


**Paper Topic And Main Contributions:**

In this paper, the authors investigate the efficacy of artificially augmenting training data for morphological inflection. Theoretically and empirically, the authors show that using synthetic data in training improves overall performance. They used an information-theoric framework with simplifying assumptions to show that adding synthetic data is useful. Empirically, they used multiple setups to sample synthetic examples from a synthetically generated dataset and evaluated the different training scenarios. The synthetic data is generated from a small sample of gold data from UniMorph. The experiments were conducted on 7 typologically different languages.

**Questions For The Authors:**

- QA: In the HighLoss (and LowLoss) sampling, it is not clear (at least to me) whether the uncertainty is counted for the whole entry (X,Y,T) or just Y.


**Reasons To Accept:**

This paper is well-written and easy to follow for the most part.
- **low-resource aid**: This work supports a technique useful for low-resource languages and varieties, especially for the morphological inflection task.
- **pushing boundaries**: The techniques described in this work will surely encourage more researchers to revisit the task of morphological (re)inflection from a different perspective, whether training or evaluating existing models.

**Reasons To Reject:**

- **lack of MSD splits**: A key evaluation for compositionality would be a split test set based on the feature tag or the Morphosyntactic descriptions (as the authors referred to it). This evaluation will highlight additional strengths or weaknesses of using synthetic data.
- **highly concatenative morphology only**: This was briefly addressed in the paper. Given that one of the test languages, i.e., Arabic, employs templatic morphology, it would be important to investigate the behavior of _STEMCORRUPT_ when generating synthetic data from entries where the inflection is a change of the stem template.

**Reproducibility:**

4: Could mostly reproduce the results, but there may be some variation because of sample variance or minor variations in their interpretation of the protocol or method.

**Reviewer Confidence:**

4: Quite sure. I tried to check the important points carefully. It's unlikely, though conceivable, that I missed something that should affect my ratings.

---

> ### Author Rebuttal · Authors · 2023-08-29
>
> We thank the reviewer for their thoughtful feedback. We’re glad that they found it well written and that they consider it to push the boundaries of morphological inflection research, especially in low-resource inflection. We also agree with their identification of the main contributions of the work. We hope that the following responses thoroughly address their reservations about the work:
>
> > lack of MSD splits: A key evaluation for compositionality would be a split test set based on the feature tag or the Morphosyntactic descriptions (as the authors referred to it). This evaluation will highlight additional strengths or weaknesses of using synthetic data.
>
> We thank the reviewer for bringing the paper by Kodner et al. (2023) to our attention. It is clearly related to the compositional generalization of morphological inflection systems; Kodner et al. demonstrate that generalizing to novel feature combinations (”featsNovel”) is an aspect of compositionality that current models struggle with. Given that it is a paper that studies compositionality in morphological inflection, it is clearly very related and we will cite it.
>
> However, we do not believe that the lack of MSD splits in our experiments is grounds for rejection, for two reasons. First, please note that this paper was very recent at the time of submission. It was posted on arxiv on **May 25th, 2023**, which was less than a month from the submission deadline and after the anonymity deadline.
>
> Second, we don’t believe experiments on the MSD split are directly relevant to the questions we ask and the conclusions we draw in our work. There are several aspects of compositionality in morphological inflection, as clearly delineated by Kodner et al. (2023). We target one important aspect. Namely, that which was highlighted by Goldman et al. (2022) and Liu & Hulden (2022), and that Kodner et al. (2023) call  the “lemmaNovel” setting, which showed that systems struggle to generalize when asked to inflect novel lexemes. We theoretically analyzed the relationship between StemCorrupt and the “lemmaNovel” setting because of experimental results reported by Liu & Hulden (2022), which empirically demonstrated that StemCorrupt is effective for generalization in the “lemmaNovel” setting.
>
> Unlike the empirically attested relationship between StemCorrupt and the “lemmaNovel” setting, there isn’t a priori justification to believe that the synthetic data generated by StemCorrupt would be an effective technique for improving generalization in the “featsNovel” setting. The StemCorrupt algorithm does not perturb, segment, or re-combine affixes in any way that would be beneficial for improving generalization in “featsNovel”.  Since our paper is centered on the StemCorrupt method and the “lemmaNovel” setting, we don’t believe it is appropriate to include an experiment on the  “featsNovel” test split without a priori justification to expect it to garner improvements — or have any effect at all.
>
> We will clarify in the paper that we address the compositional generalization issue that is highlighted by Goldman et al. (2022) and Liu & Hulden (2022), but there are outstanding, important issues in morphological inflection compositionality — namely, generalization in the “featsNovel” setting highlighted by Kodner et al. (2023) — that heretofore remain unaddressed and merit further study.
>
> > • highly concatenative morphology only: This was briefly addressed in the paper. Given that one of the test languages, i.e., Arabic, employs templatic morphology, it would be important to investigate the behavior of STEMCORRUPT when generating synthetic data from entries where the inflection is a change of the stem template.
>
> **Arabic's templatic morphology**
>
> The SIGMORPHON data collection that we draw from does indeed include Arabic, a prototypical templatic morphology-exhibiting language. Prompted by this review, we further investigated the behaviour of StemCorrupt in Arabic. We find that the StemCorrupt algorithm is only partially effective for Arabic, due to its non-concatenativity. At an aggregate level, StemCorrupt brings improvements because in most cases in the Arabic SIGMORPHON 2018 dataset, the inflection comprises a concatenative transformation (i.e., add a prefix and/or suffix; e.g., مَطَارَات → مَطَارٌ).
>
> Inflections requiring stem-internal changes however (e.g., كِتَابَة → أَكْتُبُ) are not benefited from StemCorrupt.  The stem discovery algorithm used in StemCorrupt does not isolate Arabic consonant roots for perturbation (like ktb), but rather entire stems (containing a mix of consonants and vowels, like كِتَاب). Since vowels in Arabic serve as exponents of morphological information, StemCorrupt thus has the undesirable effect of obfuscating the statistical dependency between non-root segments in the stem with the morphosyntactic description (MSD).
>
> We will add the discussion of how StemCorrupt behaves with regards to Arabic stem-changing inflections to the paper. Moreover, we will add quantitative analysis concerning the combined effect of the sampling schemas described in Section 4.2 and the inability of StemCorrupt to correctly identify consonantal roots in Arabic. This analysis will follow the structure of the vowel-harmony analysis we conducted for Turkish.
>
> **Vowel harmony analyses**
>
> It is also important to note that the paper even in its current form **does** in fact go beyond analyzing simple concatenative morphology. While we didn’t analyze the Arabic predictions in-depth specifically, we do nonetheless perform an in-depth case study on a non-concatenative morphophonological phenomenon.
>
> Specifically, we investigate vowel harmony in Turkish, where the vowels in the suffix must adhere to the frontness of the initial vowel in the stem (lines 229 - 238; 537-567). We investigated this phenomenon carefully as it is omnipresent in the Turkish dataset and since vowel harmony is a well understood example of non-concatenative morphology (see, for example, Amrhein & Sennrich, 2021, who also consider vowel harmony as a non-concatenative phenomenon).
>
> Our investigation revealed that StemCorrupt regularly generates synthetic data that violates this vowel harmony restriction and we find evidence that it tempers the overall effectiveness of StemCorrupt, in concordance with our theoretical analysis.
>
> > QA: In the HighLoss (and LowLoss) sampling, it is not clear (at least to me) whether the uncertainty is counted for the whole entry (X,Y,T) or just Y.
>
> We use the conditional predictive uncertainty, i.e., just for Y. This is consistent with other recent works that investigate data selection in NLP, e.g., Tamkin et al. (2022) and Muradoglu & Hulden (2022). We will clarify this in the paper.
>
> *References*
>
> Amrhein, C., & Sennrich, R. (2021). How Suitable Are Subword Segmentation Strategies for Translating Non-Concatenative Morphology? *Conference on Empirical Methods in Natural Language Processing*.
>
> Muradoglu, S., & Hulden, M. (2022). Eeny, meeny, miny, moe. How to choose data for morphological inflection. *Conference on Empirical Methods in Natural Language Processing*.
>
> Goldman, O., Guriel, D., & Tsarfaty, R. (2021). (Un)solving Morphological Inflection: Lemma Overlap Artificially Inflates Models’ Performance. *Annual Meeting of the Association for Computational Linguistics*.
>
> A. Tamkin, D. Nguyen, S. Deshpande, J. Mu, & N. Goodman. (2022). In *Advances in Neural Information Processing Systems* , A. H. Oh, A. Agarwal, D. Belgrave, & K. Cho (Eds.).
>
> Liu, L., & Hulden, M. (2022). Can a Transformer Pass the Wug Test? Tuning Copying Bias in Neural Morphological Inflection Models. In *Proceedings of the 60th Annual Meeting of the Association for Computational Linguistics (Volume 2: Short Papers)* (pp. 739–749). Association for Computational Linguistics.

---

### Official Review · Reviewer_xBw7 · 2023-08-07

**Soundness:** 4

**Excitement:**

4: Strong: This paper deepens the understanding of some phenomenon or lowers the barriers to an existing research direction.

**Paper Topic And Main Contributions:**

This paper analyzes how a synthetic data augmentation (STEMCORRUPT) method affects transformer-based morphological inflection models trained on golden 100 training instances + augmented data. This work formally proves the intuition that STEMCORRUPT is good for concatenative morphological processes yet has reduced performance for languages with non-concatenative morphological phenomena. The paper also explores how to improve the sample efficiency of STEMCORRUPT via improved synthetic data selection.

**Questions For The Authors:**

Why didn't you pick a larger training data set scenario?
Why analysing sample efficiency in a setting with 100 training instances is interesting? As I see no practical interest in this, maybe a theoretical argument on why this matters would be in place.

**Reasons To Accept:**

- Work provides a formal analysis of empirically used data augmentation method.
- The work is interesting and clearly written.

**Reasons To Reject:**

- Experimental part of the work involved training Transformer models on 100 samples large training data. The accuracy of the baseline models often is 0. While the merits of this work are theoretical, having such experiments makes me wonder if any of the shown results generalize beyond supper tiny data sets with no practical use.
- Analyzing sample efficiency in a setting in which all training samples can be presented to the model in a single batch seems to have little use.

**Reproducibility:**

5: Could easily reproduce the results.

**Reviewer Confidence:**

3: Pretty sure, but there's a chance I missed something. Although I have a good feel for this area in general, I did not carefully check the paper's details, e.g., the math, experimental design, or novelty.

**Typos Grammar Style And Presentation Improvements:**

If UMT - Uniform Morphological Template, what does UME mean? Shouldn't it be EMT - Empirical Morphological Template?
Line 327 - Wu et al. (2021); Liu and Hulden (2022) -> Wu et al. (2021) and Liu and Hulden (2022)
Line 306 - maybe it is worth saying "non Independent and identically distributed (IID)" to introduce the abbreviation.

---

> ### Author Rebuttal · Authors · 2023-08-29
>
> We thank the reviewer for their feedback. We are pleased to hear that they found our work interesting and clearly written, and we agree with their assessment of the main contributions of the paper. The review also demonstrates that there are some important clarifications to make about the experimental sections of our paper. We hope that these clarifications will address any reservations the reviewer may have had about our work. Please see below for our responses to the critiques and questions:
>
> > Experimental part of the work involved training Transformer models on 100 samples large training data. The accuracy of the baseline is often 0.
>
> We think there is some confusion here that is important to clarify. In our problem set up (Section 4; line 314), we introduce two training data subsets: one containing gold-standard examples ($D_{train}$) and another one containing synthetic examples generated by StemCorrupt ($\hat{D}^{Syn}_{Train}$). In the experimental part of our work, we train on $\hat{D}^{Syn}\_{train}\cup D\_{train}$. While $\lvert D\_{train}\rvert = 100$, we thoroughly evaluate performance on using $\lvert{\hat{D}}^{Syn}\_{train}\rvert\in \{128,256,512,1024,2048\}$. Our full training sets ($D\_{train} \cup \hat{D}^{Syn}\_{Train}$) can thus in fact be as large as $|D\_{train}| + |\hat{D}^{Syn}\_{Train}| = 100+2048=2148$.
>
> There is only one experimental setting in which we use only 100 training examples, where $\hat{D}^{Syn}\_{Train}=\emptyset$.  This setting is described in Section 4.1 (line 324 of the paper). This setting is not meant to serve as a competitive baseline model. Rather, it only serves to demonstrate that a Transformer trained with no data augmentation almost entirely fails to generalize. This has been shown in previous works (Liu & Hulden, 2022; Goldman et al., 2022). We don’t however necessarily expect readers to be familiar with the experimental results of those prior works so we replicate it in our study. It’s also useful for contextualizing the rest of the experimental results presented in Section 5.1. We can clarify this in the camera-ready.
>
> It is entirely possible that we induced the confusion about the size of the training dataset in lines 331-334, where we write that “we conduct all of our experiments using 100 gold-standard examples, adhering to the low-resource setting for SIGMORPHON 2018”. We intended to convey that it is the gold-standard set $D\_{train}$ — not $\hat{D}^{Syn}\_{Train}$ — that is fixed to 100 gold-standard training examples. We can certainly clarify this in the camera-ready.
>
> > Why didn't you pick a larger training data set scenario?
>
> We think that the explanation above may in part alleviate the reviewer’s concern of using small training datasets, as the training datasets are larger than the reviewer may have thought. (Up to $2,148$ examples in our experiments, rather than $100$). Further, we’re interested in the sample-efficiency of the data augmentation method StemCorrupt, not the sample-efficiency of the limited gold-standard training data.
>
> The reviewer may still ask why we only used only 100 gold-standard training examples. We want to emphasize that developing neural morphological inflection systems that use only 100 gold-standard training examples is actually entirely in adherence with precedence in experimental research on low-resource morphological inflection.
>
> Consider the seminal paper that introduced the idea of StemCorrupt (Silfverberg et al., 2017), as well as the second paper that further enhanced the method in producing state-of-the-art morphological inflection results (Anastasopolous & Neubig, 2019). Both papers were motivated by the challenge of training models that can generalize (to a reasonable degree) on extremely scarce datasets, with up to 100 gold-standard training examples. Based on this precedent, we believe our fine-grained analysis of StemCorrupt in a data-constrained setting is appropriate, as this constrained setting is when data augmentation has proven to be an essential component of the training process.
>
> Note further that the research community interested in morphology (SIGMORPHON) has had a keen interest in developing systems that work for lower-resourced languages, some of which are undergoing concerted language revitalization efforts (see Wiemerslage et al., 2022, for a review and Moeller et al., 2020 for case studies). For such languages, having only about 100 cleanly morphologically annotated word types is not at all unusual. Thus, there is highly practical interest in understanding the behaviour of data augmentation strategies in extremely data-scarce settings, which is precisely what we set out to do in the experimental section of our work.
>
> > Why analysing sample efficiency in a setting with 100 training instances is interesting?
>
> * It is important to reiterate that we’re interested in the sample efficiency of data augmentation rather than that of the gold-standard training data. The large synthetic dataset that we subsample from has $10,000$ datapoints, not $100$ (Section 5, line 422-427).
>
> * The motivation behind our experimental sections comes from the current standard of using data augmentation in morphological inflection, which is to use thousands of synthetic datapoints. Anastasopolous & Neubig (2019) used $10,000$ synthetic datapoints for obtaining their state of the art results in low-resource inflection, while the gold-standard training data only has $100$ datapoints . We think that the disparity between the size of the gold-standard training data and the synthetic data merited the type of investigation we conducted — do we really need synthetic data set that is 100x larger than the gold-standard training set? Could the models not attain the benefits of data augmentation using a synthetic dataset that scales linearly (100s of synthetic datapoints) or even logarithmically (dozens of synthetic datapoints) in the size of the gold-standard training data?
>
> * While it’s true that this is more of a theoretical concern, as training on even 10,000 word-level examples is not a particularly computationally intensive task for modern GPUs, there is intrinsic value in understanding how important data augmentation methods like StemCorrupt scale in relation to gold-standard data.
>
> > As I see no practical interest in this, maybe a theoretical argument on why this matters would be in place.
>
> It’s also worth noting that our experimental investigation reveals highly practical insights. First, it is important to avoid instances where the model is highly confident about the correct prediction (i.e., LowLoss strategy in Section 4.2). Our experiments show that  selecting subsets of synthetic data that appears “easy” to the model drastically underperforms every other strategy (Figure 4 and paragraph starting line 515).
>
> This finding may suggest that we should select synthetic examples where the model has high predictive uncertainty. Indeed, this is an approach that has been evaluated and championed in adjacent active-learning research (e.g., Tamkin et al., 2022). However, our analyses with the non-concatenative phenomenon of vowel harmony in Turkish and Finnish reveal that selecting synthetic datapoints by high predictive uncertainty is a poor strategy when the data augmentation method systematically differs from the ground-truth data generating probability distribution; see the paragraph starting on Line 537 for a deeper analysis. That is, when we pick synthetic examples by high uncertainty, we’re prone to picking examples that result in the model drawing systematically incorrect inferences.
>
> This latter argument connects to our theoretical analysis in Section 3: we must understand the underlying probabilistic model that the data augmentation method encourages. This will in turn inform our practical data subset selection strategies. Overall, we believe these are practical insights that are relevant to NLP researchers interested in limited data generalization and data augmentation at large, not only in morphological inflection.
>
> Note further that these results might not be apparent if we were to use a higher-resourced setting (with more ground-truth training examples), since in those settings data augmentation is no longer an essential component of the system. The variation in the selection strategies would likely then be rather small and would consequently require running the experiments with a substantially larger number of random seeds in order to have sufficient statistical power to detect any effect.
>
> > If UMT - Uniform Morphological Template, what does UME mean? Shouldn't it be EMT - Empirical Morphological Template?
>
> We thank the reviewer for this helpful suggestion; Empirical Morphological Template (EMT) is indeed a more appropriate name for the strategy and we will update this in the paper.
>
> > Line 327 - Wu et al. (2021); Liu and Hulden (2022) -> Wu et al. (2021) and Liu and Hulden (2022) Line 306 - maybe it is worth saying "non Independent and identically distributed (IID)" to introduce the abbreviation.
>
> We will make these updates.
>
> *References*
>
> Moeller, S., Liu, L., Yang, C., Kann, K., & Hulden, M. (2020). IGT2P: From Interlinear Glossed Texts to Paradigms. *Conference on Empirical Methods in Natural Language Processing*.
>
> Silfverberg, M., Wiemerslage, A., Liu, L., & Mao, L.J. (2017). Data Augmentation for Morphological Reinflection. *Conference on Computational Natural Language Learning*.
>
> Anastasopoulos, A., & Neubig, G. (2019). Pushing the Limits of Low-Resource Morphological Inflection. *Conference on Empirical Methods in Natural Language Processing*.
>
> A. Tamkin, D. Nguyen, S. Deshpande, J. Mu, & N. Goodman. (2022). In *Advances in Neural Information Processing Systems* , A. H. Oh, A. Agarwal, D. Belgrave, & K. Cho (Eds.).
>
> Wiemerslage, A., Silfverberg, M., Yang, C., McCarthy, A.D., Nicolai, G., Colunga, E., & Kann, K. (2022). Morphological Processing of Low-Resource Languages: Where We Are and What’s Next. *Findings*.
>
> Liu, L., & Hulden, M. (2022). Can a Transformer Pass the Wug Test? Tuning Copying Bias in Neural Morphological Inflection Models. In *Proceedings of the 60th Annual Meeting of the Association for Computational Linguistics (Volume 2: Short Papers)* (pp. 739–749). Association for Computational Linguistics.
>
> Goldman, O., Guriel, D., & Tsarfaty, R. (2021). (Un)solving Morphological Inflection: Lemma Overlap Artificially Inflates Models’ Performance. *Annual Meeting of the Association for Computational Linguistics*.

---

### Meta-Review · Area_Chair_9NUJ · 2023-09-19

**Recommendation:** 5

**Metareview:**

The paper analyses a data augmentation technique, STEMCORRUPT. The method is used to produce synthetic examples for the morphological inflection task by substituting stem characters. The authors demonstrate the utility of the technique, both theoretically and empirically. The paper also explores data-efficient strategies to sample from the synthetically generated dataset. As most reviewers outline, these research directions are particularly important for low-resource language technology. The reviewers also noted that experiments are conducted on 7 typologically diverse languages, giving this work a stronger basis. Some reviewers pointed out the flaws such as small training data and the focus on concatenative morphology only but the authors addressed and clarified them very neatly.
Overall, I think the work will be a nice contribution to NLP, and low-resource technologies in particular.

---

### Decision · Program_Chairs · 2023-10-07

**Decision:**

Accept-Main

**Comment:**

The paper analyses a data augmentation technique, STEMCORRUPT. The method is used to produce synthetic examples for the morphological inflection task by substituting stem characters. The authors demonstrate the utility of the technique, both theoretically and empirically. The paper also explores data-efficient strategies to sample from the synthetically generated dataset. As most reviewers outline, these research directions are particularly important for low-resource language technology. The reviewers also noted that experiments are conducted on 7 typologically diverse languages, giving this work a stronger basis. Some reviewers pointed out the flaws such as small training data and the focus on concatenative morphology only but the authors addressed and clarified them very neatly.
Overall, I think the work will be a nice contribution to NLP, and low-resource technologies in particular.